



**A mathematical model to improve water storage of glacial lakes prediction**
**towards addressing glacial lake outburst floods**
Miaomiao Qi[a,b], Shiyin Liu[a,b,c]*, Yongpeng Gao[d,e], Fuming Xie[a,b], Georg Veh[f], Letian Xiao[a,b],
Jinlong Jing[g], Yu Zhu[a,b], Kunpeng Wu[a,b]
[a] *Yunnan Key Laboratory of International Rivers and Transboundary Eco-Security, 650091 Yunnan*
*University, Kunming, China;*
[b] *Institute of International Rivers and Eco-Security, Yunnan University, 650091, Kunming, China;*
[c] *Yunnan International Joint Laboratory of China-Laos-Bangladesh-Myanmar Natural Resources*
*Remote Sensing Monitoring, Kunming 650091, China;*
[d] *Faculty of Geography, Yunnan Normal University, Kunming, 650500, China;*
[e] *Key Laboratory of Resources and Environmental Remote Sensing for Universities in Yunnan,*
*Kunming 650500, China;*
[f] *Institute of Environmental Science and Geography, University of Potsdam, Potsdam, Germany*
[g] *School of Mathematics and Statistics, Yunnan University, 650091, Kunming, China;*
*Corresponding author: Shiyin Liu, shiyin.liu@ynu.edu.cn;
**Abstract:** Moraine-dammed glacial lakes are vital sources of freshwater but also pose a hazard to
mountain communities if they drain in sudden glacial lake outburst floods. Accurately measuring
the water storage of these lakes is crucial to ensure sustainable use and safeguard mountain
communities downstream. However, thousands of glacial lakes still lack a robust estimate of their
water storages because bathymetric surveys in remote regions are difficult and expensive. Here we
geometrically approximate the shape and depths of moraine-dammed lakes and provide a cost-
effective model to improve lake water storage estimation. Our model uses the outline and the terrain
surrounding a glacier lake as input data, assuming a parabolic lake bottom and constant hillslope
angles. We validate our model using ten new bathymetrically surveyed glacial lakes on the Qinghai-
Tibet Plateau, and compiled data from 34 recently measured lakes. Our model overcomes the
autocorrelation issue inherent in earlier area/depth-water storage relationships and incorporates an
automated calculation process based on the topography and geometrical parameters specific to
moraine-dammed lakes. Compared to other models, our model achieved the lowest average relative
error of approximately 14% when analyzing 44 observed data, surpassing the >44% average relative
error from alternative models. Finally, the model is used to calculate the water storage change of
moraine-dammed lakes in the past 30 years in High Mountain Asia. The model has been proven to



be robust and can be utilized to update the water storage of lake water for conducting further
management of glacial lakes with the potential for outburst floods in the world.

## 1. Introduction

Moraine-dammed glacial lakes (MDLs) trap meltwater from snow and ice behind barriers of
debris at or near the termini of glaciers (Westoby et al., 2014; Yao et al., 2018; Veh et al., 2019). As
glaciers have been retreating in past decades in most mountain regions worldwide, new MDLs have
been forming, and existing ones have been growing in size and water storage (Bolch et al., 2012;
Carrivick and Tweed, 2013; Cook et al., 2018; Shugar et al., 2020; Zhang et al., 2023). During the
period from 1990 to 2018, High Mountain Asia witnessed a remarkable 52% and 54% increase in
the number and area of MDLs, respectively (Wang et al., 2020). Notably, the Eastern Himalayas
experienced the most significant growth, leading in both the number and area of MDLs during this
period. MDLs are vital water reservoirs for communities in glaciated high mountains, but were also
repeatedly sources for Glacial Lake Outburst Floods (GLOFs) (Westoby et al., 2014; Wu et al., 2019;
Gao et al., 2021; Fischer et al., 2021). According to a report by Lützow et al. (2023), a total of 630
GLOFs have been linked to MDLs occurring in 27 countries between 850 and 2022 CE. A recent
study indicates that multiple GLOFs documented from 1964 to 2022 have caused damage to
infrastructure in High Mountain Asia (Nie et al., 2023).
Compared to other dam structures, MDL's dams can be unstable and prone to sudden failure,
releasing parts of the impounded water storage in catastrophic floods (Westoby et al., 2014). MDLs
can grow towards steep slopes, where debris or ice could fall into the lakes, causing the barriers to
overflow (Emmer et al., 2014; Carrivick and Tweed, 2013; Liu et al., 2020). Due to their high
altitude and potential energy, these flood waves can attain runout distances of many tens of
kilometers, transporting and entraining large amounts of sediments from moraines and riverbanks
(Westoby et al., 2014). Many GLOFs have transformed into debris flows and their coarse debris
rapidly filled hydropower reservoirs and further destroyed infrastructure along the flow path
(Westoby et al., 2014). For example, GLOFs descending from the mountains with high kinetic
energy have recently damaged transport and power infrastructure such as the Upper Bhote Koshi
hydropower plant, with a reconstruction cost of 57 million USD (United States dollar) (Cook et al.,
2018). Future flash floods are a potential threat to major new infrastructure, for example, hundreds



more hydropower projects (Nie et al., 2023). GLOFs may also undercut hillslopes along mountain

rivers, which may fail, impound river runoff, and form potentially unstable lakes. Thus, MDLs have

become a major glacier-related hazard in high mountains, and will likely remain so as glaciers could

lose more than a third of their mass by the end of the 21st century (Rounce et al., 2023). Appraising

the water storage of glacial lakes is key to allowing for sustainable development along river channels

originating in glaciated headwaters (Yao et al., 2018; Harrison et al., 2021; Shugar et al., 2020; Liu

et al., 2020).

The peak discharge during GLOFs, a quantity commonly used to assess flood hazard

assessments, is linked to the water storage of the lake (Clague et al., 2000; Westoby et al., 2014;

Sattar et al., 2021; Nie et al., 2023). The failure of the MDLs with the largest water storage has

sustained high discharges for many hours, causing widespread inundation in mountain valleys

(Mergili et al., 2020). The Sangwang Tsho experienced disastrous outbursts in July 16, 1954,

featuring one of the highest reported flood water storages and discharges. Researchers therefore

developed numerous empirical regression equations to predict the potential peak discharge during

an outburst from a given lake water storage (Wang et al., 2018; Veh et al., 2019; Duan et al., 2023).

In any case, these predictions and simulations of peak discharge depend on accurate estimates of

lake water storage, ideally obtained through bathymetric surveys. However, measurements of lake

depth are expensive and difficult to conduct in high-altitude regions with limited access (Cook and

Quincey, 2015; Qi et al., 2022). Therefore, in situ measurements of lake depth are available only for

a few dozen cases in the Himalayas, while the water storage remains unknown for the other

thousands of lakes in this region. Current optical or radar-based satellite missions, while useful for

mapping lakes, are limited in measuring lake bathymetry due to the strong attenuation of

electromagnetic waves in glacial lakes (Zhu et al., 2019). As such, there has been an ongoing effort

to refine empirical scaling relationships from the few available worldwide samples that relate glacial

lake depth and/or area to lake water storage (Fujita et al., 2013; Loriaux and Casassa, 2013;

Carrivick and Quincey, 2014; Cook and Quincey, 2015; Veh et al., 2019; Shugar et al., 2020; Qi et

al., 2022). However, these equations may yield significant errors in orders of magnitude for a given

lake area due to the the autocorrelation issue inherent in earlier area/depth-volume relationships.

Although there are models considering the specific geometric shapes and topography around lakes

to estimate water storage of larger size plateau tectonic lake (Zhou et al., 2020; Zhu et al., 2019).





After numerous experiments, we have found that the aforementioned models do not apply to
estimating the water storage of glacier lakes due to the lack of consideration for glacial lake and
related parameters. Given the critical role of glacial lake water storage in assessing hazard risk and
providing early warning information, the development of a mathematically robust yet cost-effective
model is urgently needed.
Our goal is to introduce a novel approach for accurately estimating water storage by
incorporating its geometry and surrounding terrain. To this end, we propose a three-dimensional
model to approximate the basin morphology of MDLs and derive its analytical equation. We assess
the performance of this model against field-measured underwater topography data and further
compare the model error against other available empirical scaling relationships. Finally, we discuss
the uncertainty and rationality of the new model and apply the model to estimate the water storage
of a moraine-dammed lake in High Mountain Asia.
**2. MDLs types and their geometric approximation**
MDLs can be classified into glacier-contacted lakes (GCL) and glacier-uncontacted lakes
(GUL). GCLs are supraglacial ponds on top of debris-covered glaciers or lakes at the termini of
glaciers (Richardson 2000; Bennett et al., 2012). We term GCL as MDL in direct contact with the
glacier terminus (Figure 1a). By contrast, GULs are separated from the present glaciers, but
impound substantial parts of the meltwater from the glacier upstream (Figure 1b). The bottom of a
MDL may be a sediment-covered bedrock depression, eroded and deepened by the parent glacier
during earlier advances. As glaciers retreat, they provide space for lakes to grow between the glacier
terminus, with the abandoned moraine trapping excess meltwater from the parent glacier (Nie et al.,

2023).

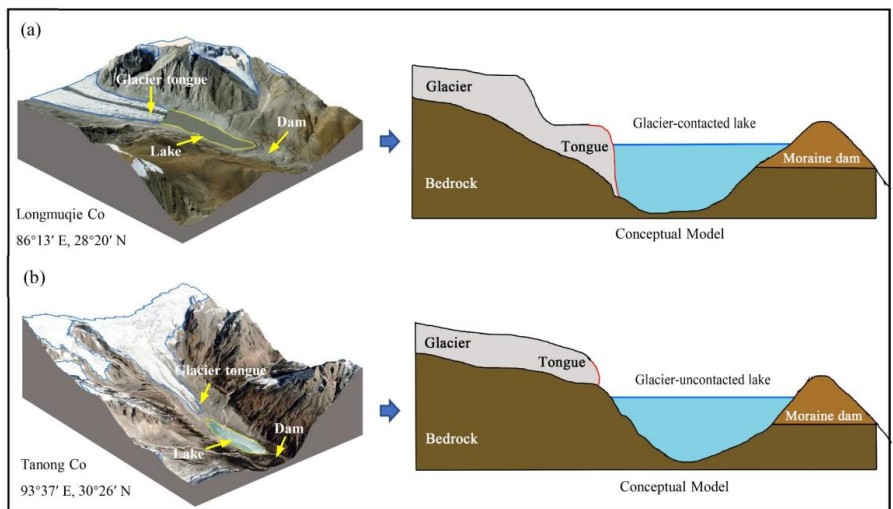

**Figure 1**. Longitudinal cross-sections along a glacier-contacted (a) and glacier-uncontacted lake (b) (The base images are from Google Earth imagery) (©Google Earth). Sketches are idealized and do not represent measured elevations.

We use the glacial lake inventory of High Mountain Asia by Wang et al. (2020) to differentiate these two types of MDLs. In general, glacial lakes grow in area largely because they become longer. Lower values of the ratio ($R$) between the maximum width and maximum length indicate that the shape of the lake is elongated; $R$ equals 1 if the lake is perfectly circular or square (Qi et al., 2022). According to the glacial lake inventory, the $R$ value for glacial lakes in High Mountain Asia ranges from 0.1 to 1.0. When $R$ is less than 0.1, it indicates the presence of glacial lakes with lengths exceeding 10 meters but widths of approximately 1 meter. However, in reality, glacial lakes with such dimensions are practically non-existent. Therefore, thresholds of $R$ allow us to distinguish glacial lakes into four subclasses (Table 1). We find that newly formed GCLs typically have small surface areas and high values of $R$. We classified GCLs with $R$ between 0.70 ~ 1.0 as GCL-1, and those with $R$ less than 0.69 as GCL-2. Examples of these two types are Poiqu No.1 Lake (85.92°E, 28.14°N) and Bienong Co (93°26′E, 30°31′N) (Table 1). With ongoing glacier recession, lakes might become decoupled from their parent glacier, switching from a lake-terminating to a land-terminating glacier. We termed lakes as GUL-1, if $R$ ranged between 0.5 and 1.0, and GUL-2 if $R <$ 0.49. Paqu Co (86°15′E, 28°30′N) and Jialong Co in 2020 are the examples of these two classes (Table 1). It is noteworthy that the establishment of the $R$ threshold in this study is grounded in the glacial lake catalog dataset developed by Wang et al, (2020). Initially, the glacial lakes were divided



into two major categories, GCL and GUL. Subsequently, $R$ values for each glacial lake were
calculated, and all co-authors classified the geometric shapes based on different types and sizes of
glacial lakes. Ultimately, through statistical analysis of glacial lake sizes for different types, we
defined the threshold for $R$. This allows the model to automatically categorize glacial lakes based
on this value.
**Table 1** Examples of glacier-contacted lake and glacier-uncontacted lake. The ratio $R$ represents the maximum width
(m) divided by the maximum length (m) of the glacial lake. The vertical scale is exaggerated.

| Type | Lake bathymetry | Model | Features | $R$ |
|---|---|---|---|---|
| GCL-1 | PoiquNo.1 of 2021 (moraine — glacier) | | A newly formed MDL typically has a small scale and is located at the glacier tongue. | $0.70 \leq R \leq 1.0$ |
| GCL-2 | Bienong Co of 2021 (moraine — glacier) | | The MDL gradually grows in the area but has not yet reached the maximum range determined by the surrounding terrain. | $0.10 \leq R \leq 0.69$ |
| GUL-1 | Paqu Co of 2020 (moraine — glacier) | | As the glacier continues to retreat, the distance between the glacier tongue and the MDL gradually increases. | $0.50 \leq R \leq 1.0$ |
| GUL-2 | Jialong Co of 2020 (moraine — glacier) | | The length of the MDL increases with time due to the continuous supply with glacier meltwater. | $0.10 \leq R \leq 0.49$ |


## 3. Model Development

### 3.1. Input data

We suggest specific geometric models for the four subclasses (Table 1) to approximate the
water storages of MDLs. Our models are fed with data from a digital elevation model (DEM) and
from the outline of a glacial lake. We used a 12.5-meter ALOS PALSAR DEM, which is freely
available from the Japan Aerospace Exploration Agency (JAXA, https://www.eorc.jaxa.jp). We test
our approach using the water storage of ten glacial lakes that we bathymetrically surveyed between
2020 and 2021. Additionally, we sourced water storage data from 34 MDLs through relevant
literature references (see Appendix A for details). The outlines of these lakes match the extent at the
time of the bathymetric survey.

**3.2. Analytical equations**

We surmise that an ideal cross-section of a MDL (Figure 2) can be partitioned into three distinct

portions, $V_1$, $V_2$, and $V_3$, representing the water storage of the lake stored adjacent to the moraine
dam, at the center of the lake, and near the glacier (or bedrock if the lake is disconnected from the
glacier). The corresponding lengths of these three portions along the maximum length of the lake
are denoted by $m$, $r$, and $n$. The lake has its maximum depth, $h_1$ and $h_2$, on either side of $r$. Points $g$
and $f$ represent the positions of a sediment layer at the lake bottom, and $a$ and $\beta$ are the slopes of
near the water surface.

The core assumptions of our geometric model can be summarized such that: 1) an MDL has a

parabolic longitudinal bottom profile with a uniform sediment layer at the bottom of the lake to keep
$h_1 = h_2$, and a parabolic cross-section $P_S$ (Figs. 2; 3); (2) the lake surface shape can be approximated
by ellipses at both ends and a rectangle in between; (3) The glacier surface and the moraine dam dip
towards the lake with the same slope ($a=\beta$).

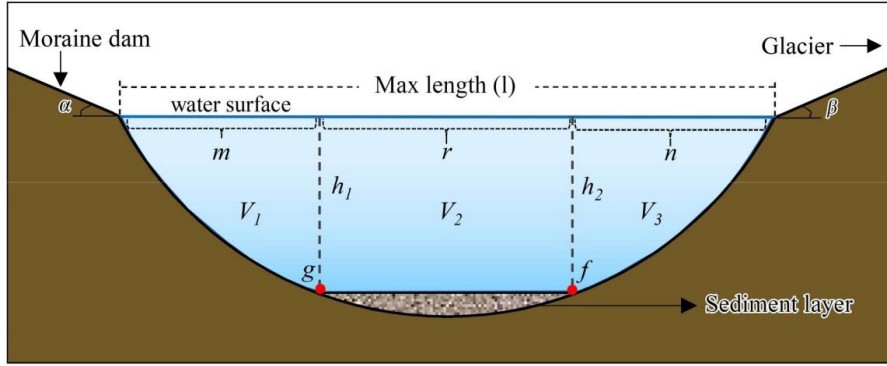


**Figure 2**. Longitudinal cross-section through a MDL. The blue horizontal line ($l$) is the maximum length on the lake
surface, subdivided by $m$, $r$, and $n$. The solid black line is the hypothetical bottom of the lake, and the gray texture
area represents a sediment layer covering the lake bottom. The maximum water depth is $h=h_1=h_2$, and points $g$ and
$f$ are at equal depths.





In three-dimensional form, the MDL basin can be divided into three parts with each having a
water storage of $V_1$, $V_2$, and $V_3$ (Figure 3a). $V_1$ and $V_3$ can be considered as the water storages of
elliptical semi-paraboloids controlled by the water depth $h$ (Figure 3b and c). Significantly, $V_1$ and
$V_3$ may or may not be equal, depending on the values of $m$ and $n$. $V_2$ is a semi-parabolic cylinder
(Figure 3d) that has height $r$, diameter $w$, and a parabolic cross-section $P_s$ (Figure 3e). Thus, the total
water storage of the MDL is $V=V_1+V_2+V_3$.

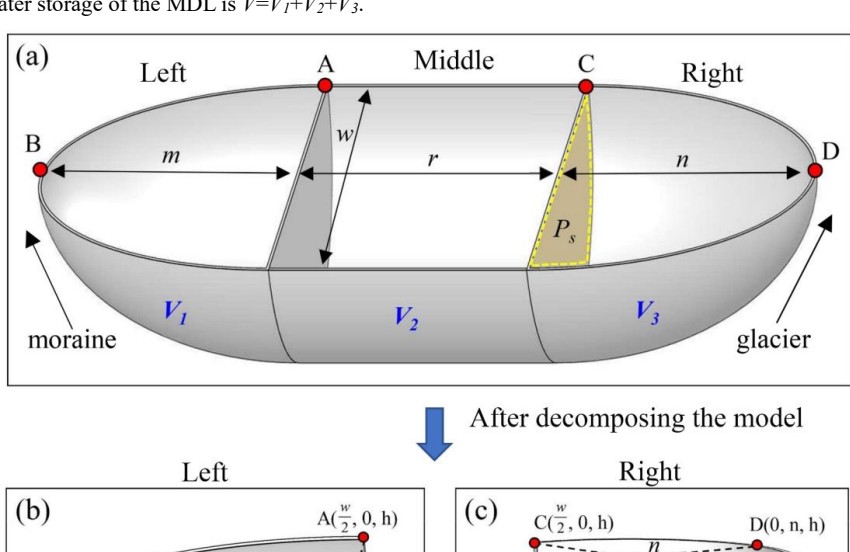

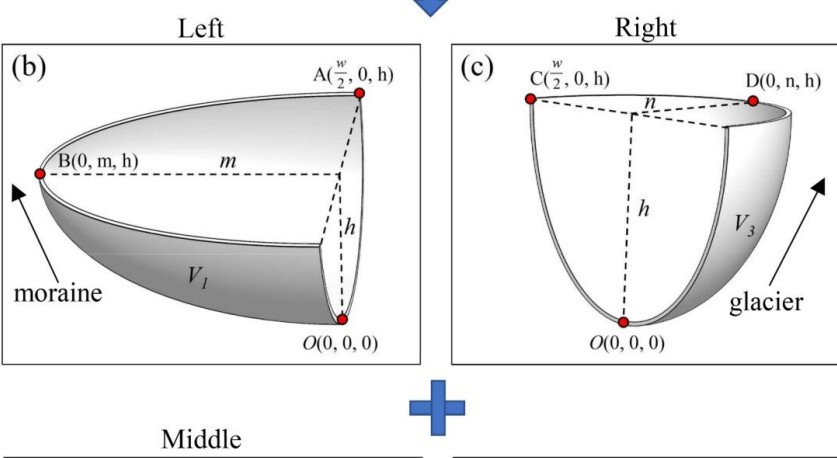

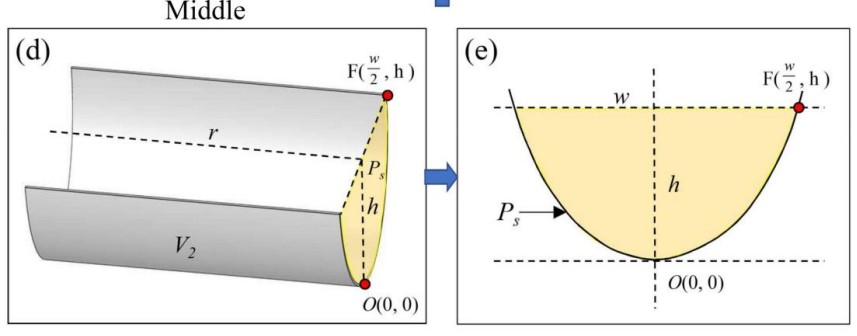






**Figure 3**. Definition diagram for the geometry of a MDL. a, hypothetical three-dimensional model of a MDL. b, Model for $V_1$ describing the lake water storage adjacent to the moraine dam. c, Model for $V_1$ describing the lake water storage adjacent to the glacier. d, Model for $V_3$ describing the lake water storage stored in the center part of the lake. e, Cross section of the column $P_s$. The parameters $m$ and $n$ are the semi-major axis of the elliptical paraboloid near the MDL inlet and outlet, respectively; $r$ is the length of the parabolic cylinder in the middle of MDL; $w$ and $l$ represent the largest width and length of the MDL, respectively; $h$ is the lake depth.

To obtain the individual lake water storages, we define the elliptical paraboloids for V1 and V2 (equations 1-2) in a Cartesian coordinate system (x, y, z) as

$$V_1 = \left\{ (x, y, z) \mid \frac{x^2}{a_1^{\,2}} + \frac{y^2}{b_1^{\,2}} \le z, y \ge 0, 0 \le z \le h \right\} \tag{1}$$

$$V_3 = \left\{ (x, y, z) \mid \frac{x^2}{a_2^{\,2}} + \frac{y^2}{b_2^{\,2}} \le z, y \ge 0, 0 \le z \le h \right\} \tag{2}$$

and the parabolic cylinder for V2 (equation 3) as

$$V_2 = \left\{ (x, y, z) \mid kx^2 \le z \le h, 0 \le y \le r \right\} \tag{3}$$

where $a_1 > 0$, $b_1 > 0$, $a_2 > 0$, $b_2 > 0$ are length of the semi-axes of upper surfaces of $V_1$ and $V_3$; $h > 0$ is the height of $V_1$, $V_2$ and $V_3$; $r > 0$ is the length of $V_2$.

Considering the four types of MDLs, GCL-1 corresponds to the case where $r=0$ and $n=0$. In this study, $m$ represents the part of the lake area closer to the moraine dam, and in most cases, $m$ is not equal to zero. However, in certain special cases, such as the Lake Zhasuo Co (93.25°E, 30.31°N) in southeastern Tibet, $m=n=0$, because the surface morphology of this lake is rectangular. In most scenarios, the water storage of the GCL-1 can be represented as:

$$V_{\text{GCL1}} = \frac{\pi wmh}{8} . \tag{4}$$

When $n=0$, the model of MDL corresponds to GCL-2, and its water storage can be represented as

$$V_{\text{GCL2}} = \frac{\pi wmh}{8} + \frac{2}{3} whr . \tag{5}$$

When $r=0$, the model of MDL conforms to GUL-1, and its water storage can be expressed as:

$$V_{\text{GUL1}} = \frac{\pi whl}{4} . \tag{6}$$

When the type of MDL corresponds to GUL-2, its water storage can be expressed as:





$$V_{\mathrm{GUL2}} = \frac{\pi w h(l-r)}{4} + \frac{2}{3} whr .\qquad(7)$$

Finally, the water depth ($h$) can be derived from the $w$ and slope angles ($a$) of the glacial lake:
$$h = \frac{w \tan(\alpha)}{4} .\qquad(8)$$

Section 1 in the Supplementary file elaborates more on the derivation of these analytical
equations, Table 2 shows the definition of the abbreviations in the model procedure.
**Table 2**. The definition of the abbreviations in the geometric model.

| Abbreviation | Description and definition |
|---|---|
| MDL | The moraine-dammed lake |
| GUL | The glacier-uncontacted lake |
| GCL | The glacier-contacted lake |
| $R$ | The ratio of the maximum width to the maximum length of the MDL |
| $m$ | The semi-major axis of the elliptical paraboloid of the MDL outlet |
| $n$ | The semi-major axis of the elliptical paraboloid at the MDL inlet |
| $c$ | The arbitrary height of the cross-section of an elliptic paraboloid |
| $r$ | The length of the parabolic cylinder in the middle of MDL |
| $h$ | The maximum water depth of MDL |
| $w$ | The diameter of the largest inscribed circle of the MDL |
| $l$ | The length of the minimum bounding rectangle of MDL |
| $P_s$ | The cross-section of the middle of MDL |
| $S_{Ps}$ | The area of the cross-section in the middle of MDL |
| $a$ | The average slope of the 80 m buffer zone around the MDL |

**3.3. Determination of model parameters**
We determined the parameters in Eq. 4 - 8, namely $w$, $l$, $a$, $m$, $n$ and $r$, using the lake boundary
and the DEM for all 44 Himalayan lakes with known bathymetry. We measured $w$ and $l$ by drawing
a minimum rectangle bounding box with length $l$ encompassing the MDL (Figure 4a). If the width
$w'$ of the bounding box of the MDL exceeds the actual width ($w$) of the lake, as in the case of the
tortuous boundary of Lake Longmuqie Co (86.23°E, 28.35°N) (Figure 4b), we assign the diameter
of the maximum inscribed circle within the MDL as $w$ in Figure 4c.



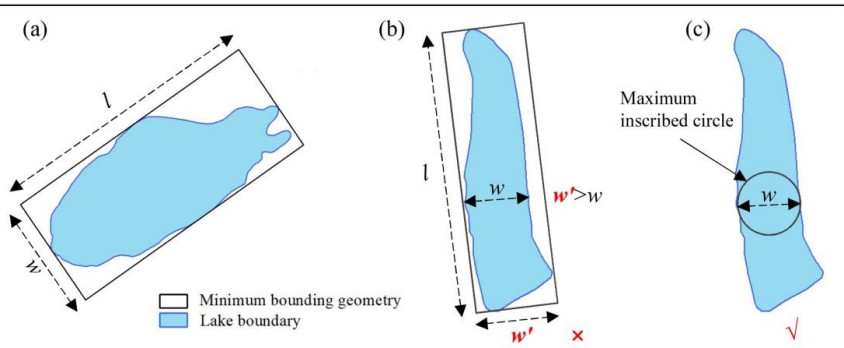

**Figure 4** Schematic illustration of the method for extracting the maximum length ($l$) and width ($w$) of the MDL. The outline in Figure a represents the geometric boundary of Lake Jialong Co (86.85°E, 28.21°N), while the outlines in Figures b and c depict the geometric boundaries of Lake Longmuqie Co (86.23°E, 28.35°N).

To determine the slope $a$-value surrounding the MDL, we use a DEM with a spatial resolution of 12.5 m in the model computation. We tested buffer sizes of 30 m, 50 m, 80 m, and 100 m width beyond the MDL boundary, and extracted the mean and median value of $a$ within each buffer. By comparing the simulated results with the measured data, we found that the water storage estimation using the median value of a within 80 m external buffer zone had a lower relative error and higher overall accuracy. Therefore, we defined $a$-value as the median slope within the 80 m buffer zone surrounding the MDL boundary. The choice of buffer zone distance can be adjusted based on the specific terrain characteristics of the research area, allowing researchers to adapt the methodology to their data accuracy.

Determining the appropriate thresholds for $m$, $n$, and $r$ of different MDL types is challenging as methods for extracting these parameters vary depending on the MDL types. In other words, due to the different types of glacial lakes, the values of $m$, $n$, and $r$ vary. Additionally, these values change with the size of the glacial lake. To enable the model to automatically identify and calculate the corresponding $m$, $n$, and $r$ for each glacial lake, we need to define a threshold. Relying on $R$, lake boundary from Wang et al. (2020) as well as DEM, $m$ and $n$ were estimated for GUL-1 and GUL-2 as shown in Table 3. In the case of GCL-1, $l = m$ due to its small area of water surface. For GCL-2, $m$ was determined as 35% of $l$ for lakes with $0.50 <R< 0.69$, 30% of $l$ for lakes with $0.30 <R< 0.49$ and 20% of $l$ for lakes with $R<0.30$ (Table 3).

For GUL-1, $R$ ranges from 0.50 to 0.10, both $m$ and $n$ are considered equal to half of $l$. On the



other hand, for GUL-2, it is possible to estimate the MDL water storage solely based on $r$, as
described in Equation 7. Accordingly, $r$ values were statistically set up as $0.4l$, $0.55l$, and $0.65l$,
respectively with three $R$ levels (Table 3). Figure 5 illustrates several representative cases of MDLs.
The above quantitative question about $m$, $n$ and $r$ is not based on subjective judgment. First,
we computed the $R$ values for all glacial lakes utilizing catalog data, then categorized them by glacial
lake type, and finally, we provided a definition by statistically assessing the shape of glacial lakes.
This definition pertains to the proportionality of $m$, $n$, and $r$ concerning the $l$ of the glacial lake.
Consequently, our model is capable of autonomously classifying each glacial lake type through
boundary data analysis. It further computes various parameters for each lake, encompassing $m$, $n$, $r$,
and h, ultimately culminating in the determination of the water storage for each lake.
**Table 3** Quantification of model input parameters.

| Lake type | Calculation rules of model input parameters | | | | | |
|---|---|---|---|---|---|---|
| | $a$ | $w$, $l$ | $R$ | $m$ | $n$ | $r$ |
| GCL-1 | | | $0.70 \leq R \leq 1.0$ | $l$ | 0 | 0 |
| GCL-2 | Median slope within the 80 m buffer zone outside the lake boundary | $w$ is the diameter of the largest inscribed circle and $l$ is the maximum length of the minimum bounding geometry | $0.50 \leq R \leq 0.69$ | $l \times 0.35$ | 0 | $l$-$m$ |
| | | | $0.30 \leq R \leq 0.49$ | $l \times 0.30$ | 0 | $l$-$m$ |
| | | | $0.10 \leq R \leq 0.29$ | $l \times 0.20$ | 0 | $l$-$m$ |
| GUL-1 | | | $0.50 \leq R \leq 1.0$ | $l \times 0.50$ | $l \times 0.50$ | 0 |
| GUL-2 | | | $0.40 \leq R \leq 0.49$ | | | $l \times 0.40$ |
| | | | $0.30 \leq R \leq 0.39$ | $l$-$r$ | | $l \times 0.55$ |
| | | | $0.10 \leq R \leq 0.29$ | | | $l \times 0.65$ |


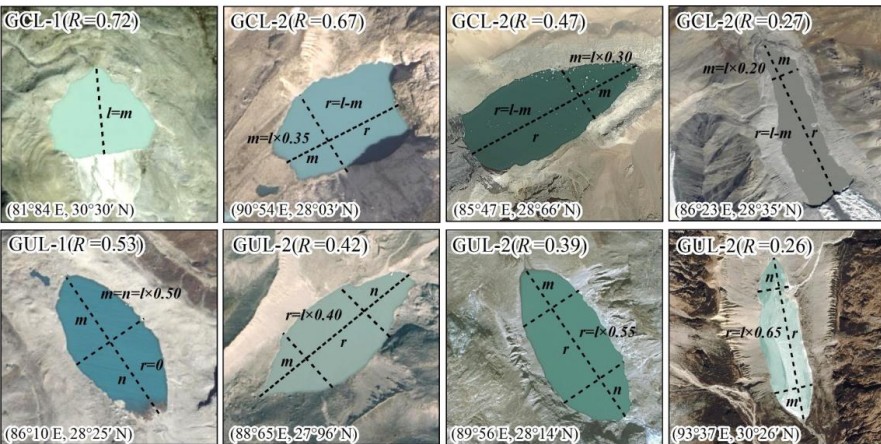


**Figure 5**. Example for the extraction of input parameters for different types of MDLs. The base map is a Google
Earth image (©Google Earth).




We trained our workflow (Figure 6) on 44 MDLs in High Mountain Asia that have known
depths and water storages. For each lake, we checked whether its outline was in contact to the parent
glacier. We automatically fitted a rectangular bounding box to calculate $R$, and then automatically
assigned each lake to one of the four types of MDL based on $R$ thresholds (Table 1). Finally, we
estimated their water storages using our and traditional empirical relationships. Our model requires
MDL boundary and DEM data as inputs, and it automatically quantifies each parameter while
selecting the optimal model for water storage estimation.
Finally, we applied our model to more than 10,000 glacial lakes with unknown bathymetry in
High Mountain Asia. This region had one of the highest rates of MDLs growth in the world in past
decades.

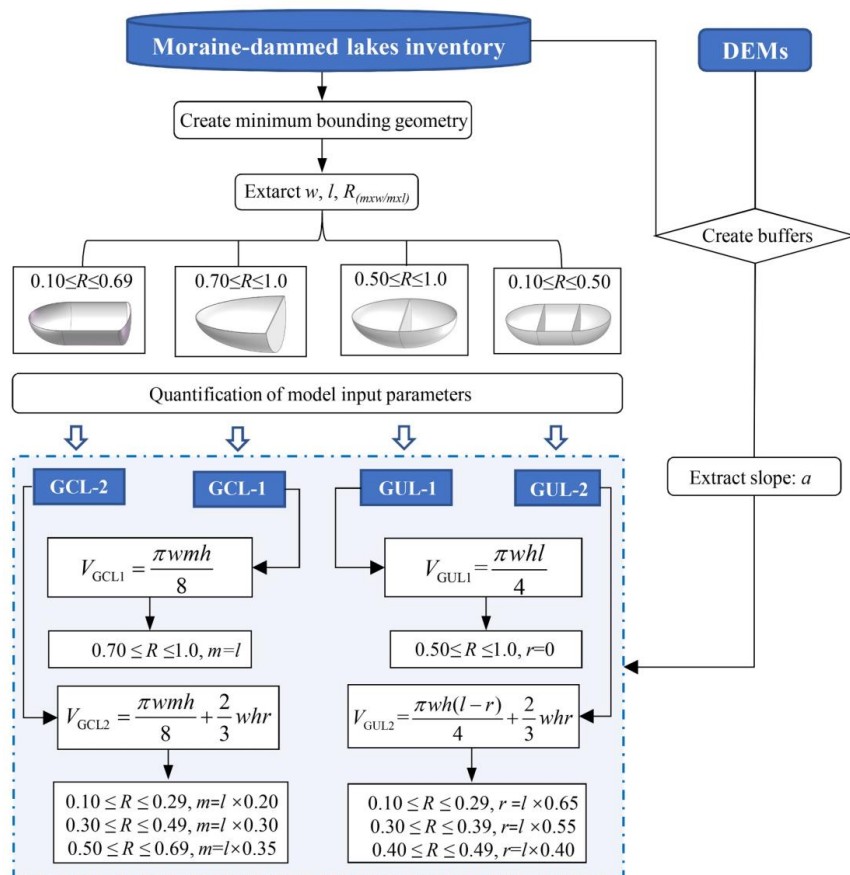


**Figure 6**. The flow chart of the model procedure derivation.



## 4. Results

### 4.1. Model validation

We validated our parameterization using bathymetry measurements from four representative glacial lakes, namely, Bienong Co, Maqiong Co, Tanong Co, and Jialong Co, located in the Qinghai-Tibet Plateau. These lakes represent the four types of glacier lakes, with depths measured through bathymetric surveying (Figure 7). In comparing estimated with measured water storages (Table 4), we find that Jialong Co has the highest accuracy with a relative error of only 1%. Maqiong Co and Tanong Co are overestimated by approximately 5% and 7%, respectively. The largest lake, Bienong Co, had an underestimated water storage of 6%.

In addition, our model is designed to approximate the mean depth of MDLs and therefore underestimates the maximum measured lake depth by about 50% (Table 4). Modeled mean water depths only deviate by 18% (mean) from the measured mean water depths. Except for a notable prediction error for Bienong Co (+47%), errors for Jialong Co, Tanong Co, and Maqiong Co range from 6% to 13% relative to the measured values.

In summary, our model has a high degree of concordance with observed glacial lake water storages and provides better estimations of water depth compared to the measured average depths. This suggests that our proposed model can used in glacial lake water storage estimation and the management of GLOF hazards.



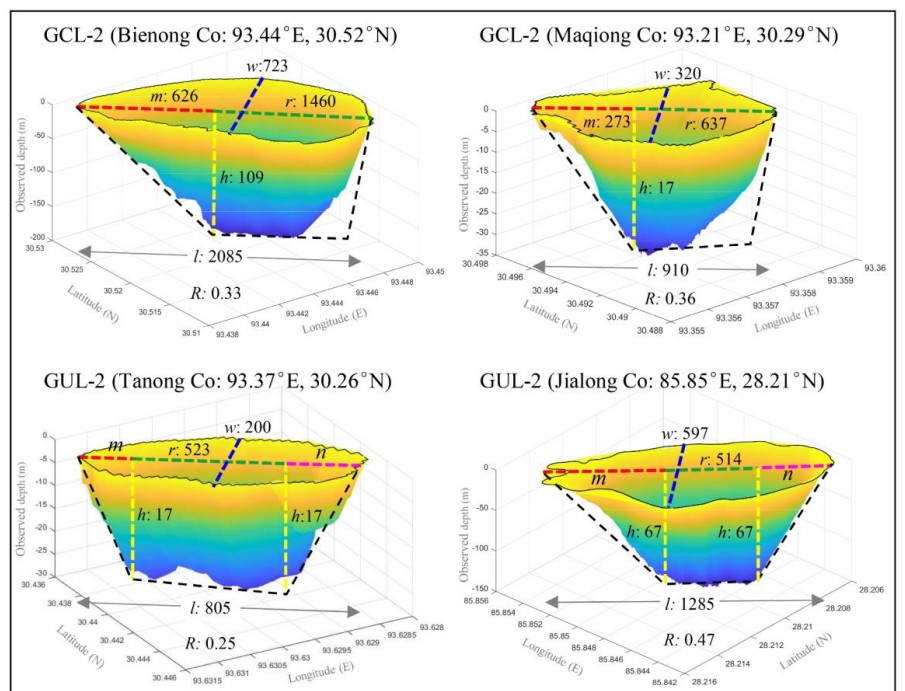


**Figure 7**. Subaqueous glacial lake morphology based on bathymetric surveys. The black dashed line represents the
hypothetical longitudinal profile of the glacial lake; $l$ and $w$ are measured from the lake boundary, $h$ is simulated
lake depth and the remaining parameters ($m$, $n$, $r$) are calculated by rule in Table 3. Lake depth is exaggerated.
**Table 4** Validation results of the mathematical model. In the column of observed values of water depth, the left
represents the maximum value and the right represents the average value)

| Name | Year of survey | Type | Area (km²) | Lake depth (max and mean, m) | | | Water storage (10⁶ m³) | | |
|---|---|---|---|---|---|---|---|---|---|
| | | | | Observed | Simulated | Error | Observed | Simulated | Error |
| Bienong Co | 2021 | GCL2 | 1.16 | 181/74 | 109 | -40/+47% | 102.000 | 95.689 | -6% |
| Maqiong Co | 2021 | GCL2 | 0.22 | 34/16 | 17 | -50/+6% | 3.325 | 3.581 | +7% |
| Tanong Co | 2021 | GUL2 | 0.13 | 29/15 | 17 | -41/+13% | 1.821 | 1.915 | +5% |
| Jialong Co | 2020 | GUL2 | 0.58 | 135/62 | 67 | -50/+8% | 37.530 | 37.952 | +1% |


**4.2. Comparison with other methods**
Table 5 displays the dataset of glacial lake bathymetry used in this study to validate the model.
We compared our model with another model that employed the lake geometry (Zhou et al., 2020),
and also with 20 additional formulas (EqS1-EqS20) collated by Qi et al. (2022) in Table S1. In the
estimation of a single MDL, formulas EqS4, EqS6, EqS13, EqS17, and EqS20 displayed significant
inaccuracies (132% - 853%). For instance, EqS13 shows an average error of 853%. Consequently,





we have refrained from conducting a comparative analysis of these five formulas in the subsequent
discussions.
**Table 5** The glacial lake bathymetry data set used in this study. The lake bathymetry data are shown in bold provided
by this study, and the rest are obtained from references, see Appendix A for details.

| Lake Name | Type | Area (km²) | Water storage(10⁶m³) | | Measurements based on remote sensing images | | | | | | |
|---|---|---|---|---|---|---|---|---|---|---|---|
| | | | Measured | Estimated | $l$ | $w$ | $R$ | $a$ | $m$ | $r$ | $h$ |
| **Kajiaqu** | **GCL2** | **0.29** | **3.45** | **3.00** | **1436** | **230** | **0.13** | **14** | **287** | **1149** | **15** |
| **Bienong Co** | **GCL2** | **1.17** | **102.00** | **95.69** | **2085** | **723** | **0.33** | **31** | **626** | **1460** | **109** |
| **Longmuqie Co** | **GCL2** | **0.58** | **8.28** | **8.47** | **1775** | **380** | **0.21** | **12** | **355** | **1420** | **21** |
| **Tanong Co** | **GUL2** | **0.13** | **1.82** | **1.92** | **805** | **200** | **0.25** | **19** | **0** | **523** | **17** |
| **Maqiong Co** | **GCL2** | **0.22** | **3.32** | **3.58** | **910** | **320** | **0.36** | **12** | **273** | **673** | **17** |
| **Zhasuo Co** | **GUL2** | **0.33** | **4.28** | **5.18** | **890** | **380** | **0.4** | **12** | **0** | **356** | **21** |
| **Jialong Co** | **GUL2** | **0.55** | **37.53** | **37.95** | **1285** | **597** | **0.46** | **24** | **0** | **514** | **67** |
| **Paqu Co** | **GUL2** | **0.58** | **8.80** | **9.22** | **2134** | **314** | **0.15** | **14** | **0** | **1387** | **19** |
| **Chmaqudan Co** | **GUL2** | **0.56** | **19.61** | **17.91** | **1459** | **450** | **0.31** | **19** | **0** | **802** | **38** |
| **Tara Co** | **GUL2** | **0.23** | **2.64** | **3.19** | **1024** | **255** | **0.26** | **15** | **0** | **666** | **17** |
| Jialong Co | GUL2 | 0.46 | 18.20 | 18.59 | 1133 | 537 | 0.47 | 17 | 0 | 453 | 41 |
| Rewuco | GCL1 | 0.42 | 13.85 | 8.52 | 839 | 613 | 0.73 | 15 | 839 | 0 | 42 |
| PoiquNo.1 | GCL2 | 0.09 | 2.53 | 2.21 | 428 | 300 | 0.64 | 22 | 150 | 278 | 30 |
| Ranzeria Co | GCL2 | 0.29 | 3.88 | 3.16 | 1181 | 288 | 0.23 | 12 | 236 | 945 | 15 |
| BethungTsho | GCL2 | 0.45 | 4.28 | 4.51 | 1355 | 373 | 0.28 | 9 | 271 | 1084 | 15 |
| Guangxie Co | GCL2 | 0.41 | 2.61 | 2.71 | 1032 | 390 | 0.3 | 7 | 310 | 722 | 12 |
| Shishapangma | GCL2 | 0.6 | 18.59 | 13.61 | 1721 | 500 | 0.29 | 12 | 344 | 1377 | 26 |
| Lugge | GCL2 | 1.63 | 71.76 | 69.02 | 3163 | 578 | 0.18 | 23 | 633 | 2531 | 62 |
| Raphstreng2 | GCL2 | 1.31 | 58.19 | 59.13 | 2117 | 816 | 0.39 | 16 | 635 | 1482 | 59 |
| Galong Co | GCL2 | 5.49 | 377.39 | 403.18 | 4284 | 1500 | 0.35 | 16 | 1285 | 2999 | 107 |
| Bnecuoguo Co | GUL1 | 0.11 | 1.69 | 1.98 | 490 | 288 | 0.59 | 14 | 0 | 0 | 18 |
| Cirenma Co | GUL2 | 0.33 | 12.43 | 12.03 | 1276 | 367 | 0.29 | 22 | 0 | 829 | 36 |
| Longbasaba | GCL2 | 1.15 | 56.16 | 43.47 | 2114 | 680 | 0.3 | 17 | 634 | 1479 | 52 |
| Midui | GCL2 | 0.22 | 1.13 | 1.34 | 968 | 280 | 0.31 | 7 | 290 | 678 | 8 |
| Lugge | GCL2 | 1.18 | 58.30 | 39.18 | 2520 | 545 | 0.2 | 19 | 504 | 2016 | 47 |
| Thulagi | GCL2 | 0.76 | 31.80 | 30.33 | 1991 | 437 | 0.22 | 28 | 398 | 1593 | 57 |
| Tsho_Rolpa | GCL2 | 1.39 | 76.60 | 62.59 | 2942 | 590 | 0.2 | 22 | 588 | 2353 | 59 |
| Imja Tsho | GCL2 | 0.6 | 28.00 | 23.18 | 1341 | 543 | 0.38 | 22 | 402 | 939 | 54 |
| Cirenma Co | GUL2 | 0.33 | 13.90 | 12.23 | 1276 | 370 | 0.29 | 22 | 0 | 829 | 37 |
| Pidahu | GCL2 | 0.89 | 50.44 | 31.37 | 2071 | 500 | 0.21 | 22 | 414 | 1657 | 50 |
| Imja Tsho | GCL2 | 1.14 | 63.80 | 52.55 | 2191 | 605 | 0.24 | 23 | 438 | 1753 | 65 |
| South Lhonak | GCL2 | 1.31 | 65.80 | 71.22 | 2328 | 715 | 0.31 | 22 | 699 | 1630 | 73 |
| Tam_Pokhari | GCL2 | 0.45 | 21.25 | 26.02 | 1178 | 470 | 0.41 | 34 | 353 | 825 | 80 |
| Thulagi | GCL2 | 0.91 | 23.30 | 31.83 | 2522 | 417 | 0.17 | 25 | 504 | 2017 | 49 |
| Imja Tsho | GCL2 | 1.03 | 35.50 | 37.03 | 2028 | 556 | 0.27 | 21 | 406 | 1622 | 54 |
| Thulagi | GCL2 | 0.94 | 35.37 | 36.19 | 2541 | 430 | 0.17 | 27 | 508 | 2033 | 54 |



| Tsho_Rolpa | GCL2 | 1.54 | 85.94 | 68.58 | 3304 | 566 | 0.17 | 23 | 661 | 2643 | 60 |
| Thulagi | GCL2 | 0.92 | 36.10 | 37.75 | 2504 | 439 | 0.18 | 27 | 501 | 2003 | 56 |
| Lower_Barun | GCL2 | 2.14 | 103.60 | 111.38 | 3297 | 730 | 0.22 | 23 | 659 | 2638 | 76 |
| Lower_Barun | GCL2 | 1.77 | 112.30 | 97.45 | 3091 | 717 | 0.23 | 22 | 618 | 2473 | 72 |
| Imja Tsho | GCL2 | 1.15 | 78.40 | 59.12 | 2208 | 610 | 0.24 | 25 | 442 | 1767 | 72 |
| Amphulapche | GUL1 | 0.12 | 3.20 | 3.79 | 404 | 369 | 0.99 | 19 | 0 | 0 | 32 |
| Chamlang Tsho | GCL2 | 0.76 | 35.00 | 26.53 | 1627 | 588 | 0.32 | 18 | 488 | 1139 | 47 |
| Imja Tsho | GCL2 | 0.75 | 33.48 | 24.13 | 1557 | 550 | 0.32 | 19 | 467 | 1090 | 48 |


Our assessment (Table 6) involves the relative error (RE, absolute value), bias, root mean
square error (RMSE), mean absolute percentage error (MAPE) and mean absolute error (MAE) to
quantify the uncertainty of new model. We use the coefficient of determination $R^2$ to describe the
goodness of fit between the model-derived data series and the measured data. Accordingly, our
model had an $R^2$ value of approximately 0.98, indicating a strong correlation between observed and
predicted lake water storages (Figure 8). Moreover, our model has the lowest variance, according
to a bias (-0.0031 $km^3$), MAE (0.0059 $km^3$), RMSE (0.0096 $km^3$), and MAPE(25%). Also, our
model has the lowest average relative error, at around 14%. The average relative error of EqS2,
EqS3, EqS5, EqS7, EqS9, EqS11, EqS15 and EqS16 ranged from 44% to 50%, while the remaining
formulas display average relative errors exceeding 50%. Although all equations achieved $R^2$ >0.93,
the predicted values have a high variance and tend to either overestimate or underestimate the water
storage of glacial lakes. Compared with our method, their bias, MAE, RMSE, and MAPE were all
55%, 64%, 52% and 64%, respectively, and thus higher than ours. EqS7 had a better prediction
accuracy. However, its bias, MAE and RMSE values are 82%, 64% and 52% higher than those of
our model, respectively. This indicates a significant estimation error for specific glacial lakes, and
both RMSE and MAE are sensitive to outliers. Overall, most of the equations tend to underestimate
glacial lake water storages, with the underestimation becoming more pronounced for larger water
storages. Nevertheless, we consider the accuracy level of our method to be acceptable due to the
lower uncertainty compared to other models, providing an alternative for predicting the water
storage of MDLs.



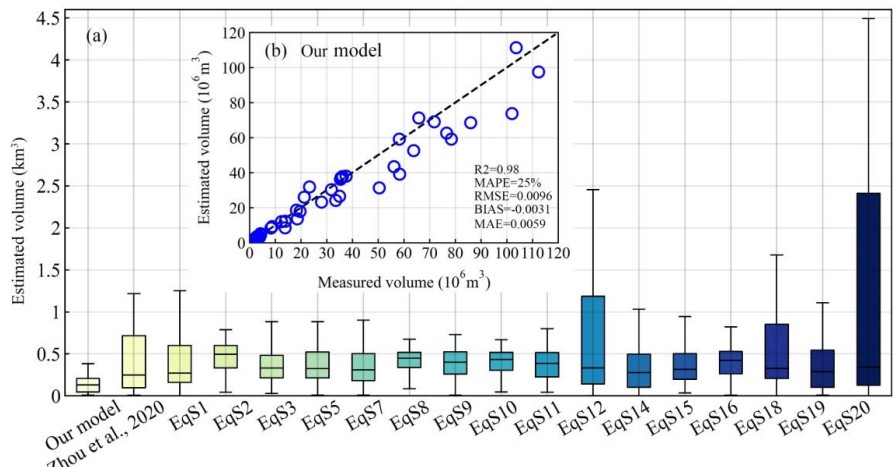


**Figure 8.** Comparison of the overall performance in glacial lake water storage estimation between our and
previous models (a) and comparison of measured and estimated water storage by our model (b).

**Table 6** Comparison of all empirical scaling relationships (EqS1-EqS20) in terms of bias, mean absolute error (MAE)
and root mean square error (RMSE) are measured in cubic kilometers. See Appendix B for details.

| Equation | RE | BIAS | MAE | MAPE | $R^2$ | RMSE |
|---|---|---|---|---|---|---|
| **Our model** | **14%** | **-0.0031** | **0.0059** | **25%** | **0.9793** | **0.0096** |
| Zhou et al., 2021 | 53% | 0.0097 | 0.0142 | 95% | 0.9289 | 0.0485 |
| Eq1 | 63% | -0.0060 | 0.0104 | 49% | 0.9654 | 0.0174 |
| Eq2 | 49% | -0.0185 | 0.0192 | 130% | 0.9521 | 0.0299 |
| Eq3 | 50% | -0.0074 | 0.0100 | 44% | 0.9556 | 0.0150 |
| Eq4 | 164% | 0.0448 | 0.0448 | 120% | 0.9494 | 0.1035 |
| Eq5 | 45% | -0.0056 | 0.0112 | 51% | 0.9418 | 0.0182 |
| Eq6 | 219% | 0.0609 | 0.0609 | 130% | 0.9509 | 0.1331 |
| Eq7 | 48% | -0.0056 | 0.0097 | 41% | 0.9516 | 0.0146 |
| Eq8 | 52% | -0.0162 | 0.0177 | 117% | 0.9621 | 0.0295 |
| Eq9 | 49% | -0.0126 | 0.0143 | 74% | 0.9556 | 0.0213 |
| Eq10 | 50% | -0.0149 | 0.0164 | 98% | 0.9596 | 0.0262 |
| Eq11 | 49% | -0.0112 | 0.0131 | 63% | 0.9551 | 0.0192 |
| Eq12 | 94% | 0.0089 | 0.0118 | 37% | 0.9642 | 0.0186 |
| Eq13 | 853% | 0.2362 | 0.2362 | 159% | 0.9590 | 0.4404 |
| Eq14 | 51% | 0.0022 | 0.0113 | 61% | 0.9438 | 0.0268 |
| Eq15 | 46% | -0.0048 | 0.0110 | 50% | 0.9430 | 0.0182 |
| Eq16 | 44% | -0.0153 | 0.0160 | 88% | 0.9288 | 0.0230 |
| Eq17 | 316% | 0.2088 | 0.2089 | 292% | 0.8736 | 0.7300 |
| Eq18 | 77% | 0.0178 | 0.0207 | 98% | 0.9418 | 0.0582 |
| Eq19 | 50% | 0.0036 | 0.0124 | 74% | 0.9379 | 0.0336 |





| | Eq20 | 132% | 0.000238 | 0.0132 | 59% | 0.9501 | 0.0245 |
|---|---|---|---|---|---|---|---|


### 4.3 Application of the new model

Considering the frequent occurrence of GLOF events in High Mountain Asia, posing threats to
downstream infrastructure and the safety of the lives and properties of the local communities,
assessing the water storage of glacial lakes is crucial for management potentially hazardous ones
(Nie et al., 2023). Therefore, this study employs a newly developed model to provide preliminary
estimates of glacial lake water storages in the study area.
A glacial lake inventory data (Wang et al., 2020) reveals that in 2018, there were a total of
13,166 glacial lakes ($\geq 0.01$ km$^2$) distributed in High Mountain Asia. The dataset highlights a
significant increase in both the number and area of GCLs from 1990 to 2018, experiencing a
remarkable growth of 52% and 54%, respectively. Model estimation results indicate that the total
glacial lake water storage in the study area was 37.18 km$^3$ in 2018. Over the past three decades, the
overall glacial lake water storage increased by 8.94 km$^3$ from 28.24 km$^3$ in 1990, representing a
growth of approximately 32%. The expansion rates of glacial lakes varied significantly across
different regions (Figure 9). Notably, the Hindu Kush-Karakoram and the central and eastern of the
Himalayas to the Hengduan Mountains witnessed the fastest increases in both glacial lake area and
water storage.

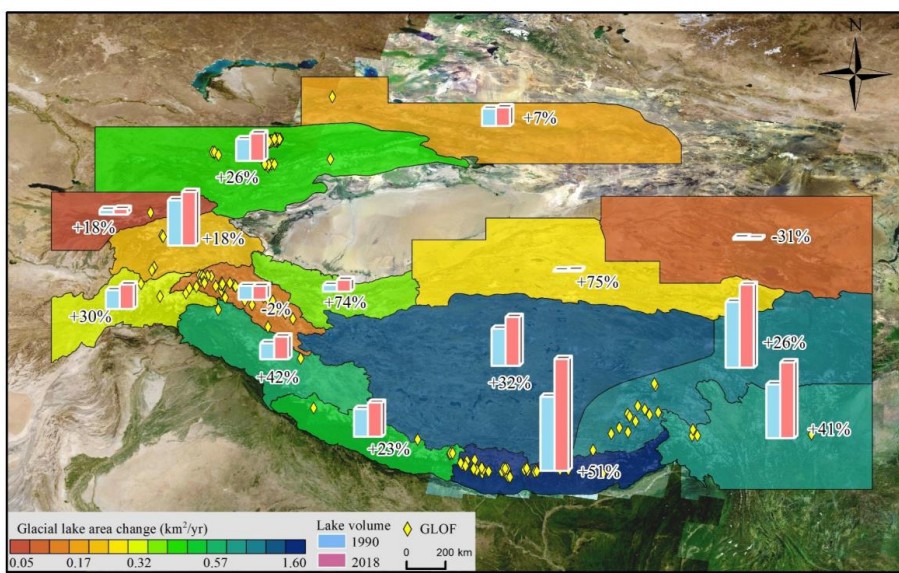


**Figure 9** Changes in the area and water storage of glacial lakes from 1990 to 2018 in High Mountain Asia. The base





map is a Google Earth image (©Google Earth).

The Eastern Himalayas had the largest gain in both the area and water storage of glacial lakes,

concurrently establishing it as a hotspot for frequent GLOFs (Figure 9). The results indicate that the
water storage of 1,410 MDLs ($\geq$0.01 km$^2$) within the study area was $9,337 \pm 990 \times 10^6$ m$^3$ in 2022.
Among these, GCLs and GULs account for 70% and 30% of the total water storage, respectively.
Between 1990 and 2022, the total water storage in glacial lakes representing a substantial growth of
162%. Notably, GCLs contributed 134% with an average annual growth rate of 8.8% a$^{-1}$, indicating
an overall increase of 280%. In contrast, the change in the water storage of unconnected lakes
remained relatively stable, experiencing a modest growth of 52% over the past 32 years,
considerably lower than that of GCLs.

At least 88 MDLs had caused 122 lake outburst floods in this area before 2022 (Veh et al., 2019,

2022; Zheng et al., 2021a) (Figure 10a), constituting approximately 44% of the total GLOF count
in High Mountain Asia. Zheng et al. (2021a) identified 280 MDLs within the study area with
extremely high potential for outburst floods. Our model suggests that although the number of MDLs
with a higher risk of outbursts is less than one-fifth of the total, their total water storage in 2022
exceeds 60% of the total water storage of MDLs in the study area. Furthermore, from 1990 to 2022,
the total water storage of these high-risk MDLs increased from $2,019 \pm 469 \times 10^6$ m$^3$ to $5,622 \pm 596$
$\times 10^6$ m$^3$, representing a substantial growth of 178%, with an annual expansion rate of approximately
5.6%·a$^{-1}$. This result is valuable as it enables practitioners to prioritize and focus their attention on
areas where the largest flood water storages are expected.
**5. Discussion**
**5.1 Justification and uncertainty of model assumptions**

In this study, we discuss the rationality and uncertainty of the model from three aspects. We

first assumed that the MDL features a parabolic longitudinal bottom profile and a uniformly
distributed sediment layer. The basin morphology of glacial lakes is a result of glacial erosion during
the glacier retreat process. Glacier erosion involves certain lateral shear stress, leading to the
formation of U-shaped valleys. Glacial lakes develop on these U-shaped valley terrains (Seddik et
al., 2009). Therefore, based on the lake bathymetry and the longitudinal bottom profile of the MDLs
(Figure 10), the variations in the underwater morphology of MDLs can be fitted with a parabolic
curve. However, when observing trends in underwater topography, it is evident that some large and



deep lakes (depth >100 m), such as Jialong Co and Bienong Co, exhibit relatively flat underwater terrain, while others do not (Figure 7). This finding aligns with the research conducted by Carrivick and Tweed (2013), who proposed that most proglacial lake basins have flat landforms resulting from extensive sedimentation. These flat terrains, which were previously subdued and smoothed by glaciation, can become covered and obscured by thin layers of silts and clays. Furthermore, it has been suggested by some scholars that in large and deep proglacial lakes, the instability of the glacier margin and the increased likelihood of wave erosion can lead to the erosion of moraine ridges at the lake bottom (Murton et al., 2012).

The underwater landforms of some MDLs are not always completely flat. As depicted in Figure 11, the bottom topography of most glacial lakes exhibits a fluctuating parabolic trend. Golledge (2008) and Bennett et al. (2000) revealed that subaqueous moraines in glacial lakes often have linear or sinuous crests, and their ridges frequently exhibit heavily glacitectonized sediment structures indicative of compression. Although the presence of subaqueous moraines is uncertain, this perspective offers a plausible explanation for the fluctuations in underwater topography. In conclusion, concerning the formation process of subglacial geomorphology in MDLs and lake bathymetry, both aspects substantiate our postulation that the MDL features a parabolic longitudinal bottom profile. Furthermore, we hypothesize the presence of uniform sediment surface to keep $h_1 = h_2$, although sediment distribution may be non-uniform due to factors such as the position of the ice margin and water density (Carrivick and Tweed, 2013). As a result, the uneven terrain at the bottom of some glacial lakes or the non-uniform distribution of sediments therein constitutes one of the sources of uncertainty in the model.

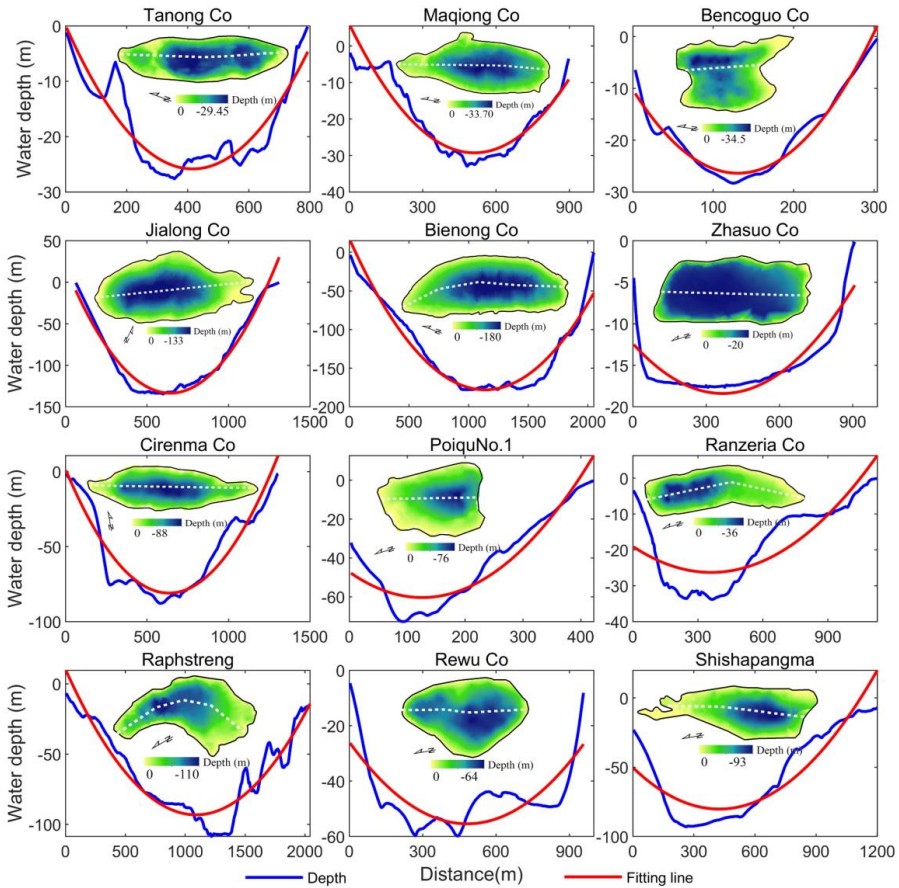

**Figure 10.** The longitudinal bottom profile underwater topography of the MDLs obtained by bathymetry and the fitting lines of terrain change trend (The white dotted line is the longitudinal profile line of the lake).

The second source of uncertainty in the model arises from the assumption regarding the lake surface of the MDL. Here, we assumed MDL's surface shape is characterized by an ellipse at both ends and a rectangle in between. MDLs develop on parabolic or U-shaped glacial troughs. A mature MDL, characterized by a relatively stable surface morphology, tends to exhibit an elliptical shape due to its geological characteristics (e.g., GUL lake type in Figure 5). Similar trends in the boundaries of MDLs are observed in different lake catalog datasets. Furthermore, in this study, MDLs are classified into four types based on their geometric shapes (see Table 1). Treating the complete geometric shape of an MDL as an ellipse allows the model to automatically partition the lake basin structure (e.g., $V_1$, $V_2$, $V_3$ in Figure 2) based on the lake's shape coefficient, facilitating the calculation of the water storage for MDLs with different morphologies. However, in reality, as





suggested by Teller (1987) and Rubensdotter et al. (2009), factors such as the position of the glacier
margin, surrounding landscape elevation and topography, and the location and elevation of lake
overflow channels can affect the basin morphology of MDLs. For instance, Bencoguo Co and
Raphstreng in Figure 10 do not exhibit the characteristic elliptical shape on the lake surface. This
uncertainty in the geometric shape of the lakes may lead to an overestimation of lake water storage
in the model, as the maximum width of the lake significantly influences the model results.
Finally, assuming the slope angle near the lake remains constant ($a=\beta$) is another aspect
contributing to the uncertainty in the model. In actuality, the slopes surrounding the lake exhibit
variations influenced by factors like the glacier tongue's position, the surrounding topography, and
the presence of moraine ridges. This variability in slope angles can further contribute to the
uncertainty when estimating the model's maximum water depth and water storage.
**5.2 Sensitivity of model input parameters**
Additionally, MDLVM requires key parameters, namely, $w$, $l$, $a$, $m$, $n$, and $r$, with the
relationship between $m$, $n$, $r$, and $l$ defined as $l = m + n + r$. Thus, we only investigated the sensitivity
of MDLVM to $l$, $w$, and $a$. Since water depth is closely related to $w$ and $a$ (see equation (13)), we
also conducted parameter sensitivity tests on the estimated water depth using MDLVM. In this study,
we employed Jialong Co and Bienong Co as representatives of GUL and GCL of MDLs, respectively,
to assess the sensitivity of the model to various parameters across different types of glacial lakes.
Figure 11 (a-f) demonstrates the sensitivity of volume ($v$) and water depth ($h$) in MDLVM to
variations in the maximum length ($l$), maximum width ($w$), and slope ($a$) of glacial lakes. Overall,
there was a linear increase in glacial lake volume with changes in length (Figures a and d). As shown
in Figures 11b and e, variations in maximum width exhibited a consistent power-law relationship
with volume, where volume increased exponentially with width. The water depth of glacial lakes
demonstrated a linear increase with changes in width. The slope of the lake's edge showed a power-
law relationship with both estimated water depth and volume (Figures 11e and f). In summary, when
estimating volume using MDLVM, glacial lake width and slope were found to be the most sensitive
parameters, followed by the lake's length. Regarding water depth, the model was most sensitive to
the slope, followed by the width.

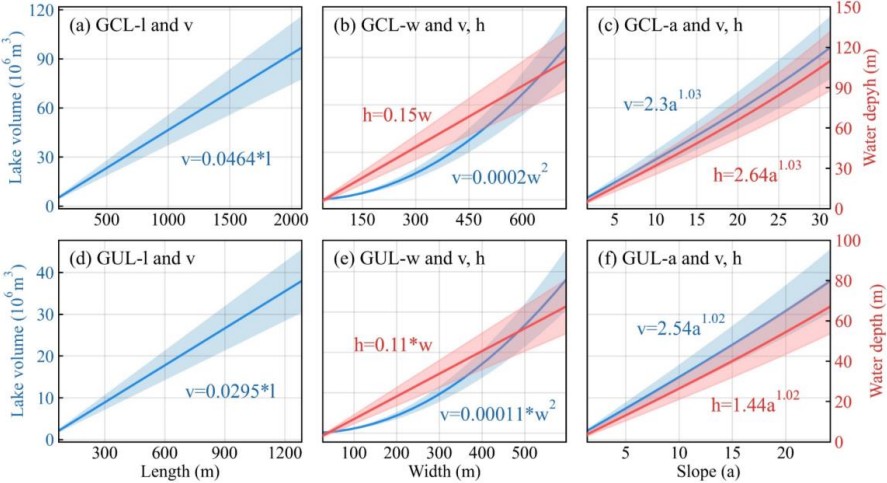


**Figure 11**. Parameter sensitivity analysis for glacial lake volume estimation using new model (note: the shaded
part represents the confidence interval, and definition of parameters in the figure as shown in Table 2).
**6. Conclusion**
Water storage plays a crucial role in predicting outburst water storage and peak discharge of
GLOFs. This study proposed a mathematically robust and cost-effective approach for estimating
lake water storage in regions where field measurements of bathymetry are limited. The new model
utilized lake geometry and DEMs to estimate lake water storage. By parameterizing the model based
on assumptions such as a parabolic longitudinal bottom profile and consistent slope angles, it offers
a reliable estimation of lake water storage.
We validated our parameterization using bathymetry measurements from four representative
glacial lakes, namely, Bienong Co, Maqiong Co, Tanong Co, and Jialong Co, located in the Qinghai-
Tibet Plateau. Additionally, we applied the new model to 10 glacial lakes with depth measurements
conducted during 2020-2021, and we included bathymetry data from 34 other glacial lakes sourced
from published literature. Our model overcomes the autocorrelation issue inherent in earlier
area/depth-water storage relationships and incorporates an automated calculation process based on
the topography and geometrical parameters specific to MDLs. Compared to other models, our model
achieved the lowest average relative error of approximately 14% when analyzing 44 observed data,
surpassing the >44% average relative error from alternative models. This study model will allow
researchers and practitioners to better predict potential outburst water storages and peak discharge
of MDLs.




**Competing interests**

The contact author has declared that none of the authors has any competing interests.

**Data availability**

All data used in this study can be found in Table 5 and supplementary files.

**Acknowledgments**

This work was supported by research grants from the Second Tibetan Plateau Scientific Expedition and Research (STEP, Grant No. 2019QZKK0208), the National Key Research and Development Program of China (No. 2021YFE0116800), the postdoctoral research start-up project of Yunnan Normal University (Grant No. 013002050205503329), the National Natural Science Foundation of China (No. 42171129, 42301154).

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
