# Peer review of "A mathematical model to improve water storage of glacial lakes prediction towards addressing glacial lake outburst floods"

_Hydrology and Earth System Sciences, 2024_

## Author Comment (AC4)

The article "A mathematical model to improve water storage of glacial lakes prediction towards addressing glacial lake outburst floods" provides a simple method to estimate Moraine Dam glacier lake volume estimations based on data that can easily be obtained from permanently updated global datasets.

Major comments,

The model calculates the volume for an ideal case when the lake is symmetrical. Although unrealistic, it is a considerable improvement from the empirical equations that look into the area of the lake to estimate the volume. However, this equation relies on the right selection of r, m and n. How confident are you in the estimation of these parameters? I appreciate that the authors provide simple relationships, but I am concerned with the potential overfitting of the empirical thresholds. According to the validation section, they used 40 out of 44 lakes to come up with the thresholds and then validated the method using the 4 lakes that they left behind. Can the authours elavorated on this calibration/validation strategy?

**Explanation and revision:** Thank you very much for your affirmation and questions. We first simplified the model into four equations, with their solutions all dependent on the correct selection of $m$, $n$, and $r$. Based on the geometry of the glacial lake, we established a proportional relationship between $m$, $n$, $r$, and the glacier lake length ($l$). This proportional relationship is empirically defined and essentially represents a geometric segmentation of the glacial lake. The lake is divided into three sections, and the volume of each section is calculated separately. The total water storage of the lake is then obtained by summing the volumes of these three sections.

Therefore, we first used measured data from four glacial lakes to validate whether this proportional relationship was appropriate. After validation, we found that the empirically derived proportional relationship performed well. Hence, this study adopts this proportional relationship as the standard for the model's input parameters. No calibration or adjustments were made during this process. We have added the following explanation in lines 243 to 248 of the original text: " Based on the geometry of the glacial lake, we established a proportional relationship between m, n, r, and the glacier lake length (l). This proportional relationship is empirically defined and essentially represents a geometric segmentation of the glacial lake. The lake is divided into three sections, and the volume of each section is calculated separately. The total water storage of the lake is then obtained by summing the volumes of these three sections."

Regarding the validation section you mentioned, my approach is as follows: First, I use the data from four moraine-dammed lakes measured by our team in previous years to validate the accuracy of the model's input parameters and estimation results. Then, I collected post-1990 glacier lake measurement data (44 lakes) as a sample set to compare and validate the estimation accuracy of our model against other published

methods. Therefore, Section 4.1 of the paper focuses on model validation, while Section 4.2 compares our model with other approaches. Therefore, the threshold values for the model's input parameters were not determined from measured data but were primarily based on the segmentation of the glacial lake's geometric shape. All measured data in this study were used solely for model validation and to compare the accuracy of our model with other methods.

The previous comment applies to Figure 8 and Table 6 as well. If the available equations are compared in the same lakes where this model was calibrated, the exercise is biased, and it should be compared with independent data.

**Explanation and revision:** Thank you very much for your suggestions. There is no parameter that needs to be trained or calibrated, all parameters can be measured through the glacial lakes and their surrounding topology. Regarding the validation section you mentioned, I did not clearly explain the entire rationale and process for model validation. Therefore, based on the suggestions from all reviewers, I have added a new subsection in the methodology section: 3.4. Model validation and application.

**3.4. Model validation and application**

In this study, we initially validated our parameterization using bathymetric measurements from four representative glacial lakes surveyed between 2020 and 2021. Subsequently, we combined the data from these four lakes with the remaining six glacier lakes we measured, along with water storage data from 34 MDLs obtained from relevant literature sources (see Appendix A for details). This resulted in a dataset of 44 lakes, which was used to compare and validate the performance of our model against other existing methods.

A glacier lake inventory of the High Mountain Asia region, published by Wang et al, 2020 was used as input data for the model application to assess the water storage of moraine-dammed lakes in this region. Notably, Wang's glacier lake inventory provides a detailed classification of GCL and GUL, which has been internationally recognized. It is important to note that in his dataset, GUL refers specifically to glacier lakes that do not contact glaciers, which may not necessarily all be moraine-dammed lakes. We conducted a thorough review and made revision to ensure that we retained only those GULs classified as moraine-dammed lake.

Also, since there are four types of lakes (GCL-1, GCL-2, GUL-1, and GUL-2), the comparison should be shown by type of lake.

**Explanation and revision:** Thank you very much for your suggestion. However, after careful consideration, we did not differentiate between glacier lake types in the revised manuscript. The primary reason is that in our model, moraine-dammed lakes are subdivided into four categories to improve estimation accuracy, whereas other methods do not classify lake types. Therefore, further subdividing the results for comparison when estimating water storage would be meaningless.

Minor comments.

How do you standardize R to make it comparable to other glaciers? Does it provide an indication of potential growth through elongation, for example? For example, in line 129, when you mentioned "newly formed," does it mean that it has the potential to grow at a higher rate? In that case, how do you define new?

**Explanation and revision:** Your question is very interesting. First, regarding the standardization of R, we mention in lines 124-129 of the paper that R represents the ratio of the maximum width to the maximum length of a moraine-dammed lake. Therefore, the value of R ranges between 0 and 1, as the width of a glacier lake is always less than its length (see Figure 4). The definitions of glacier lake length and width are provided in lines 220-225. Hence, the R values of all glacier lakes can be directly compared.

Based on glacier lake inventory data, high-resolution remote sensing imagery, and R-values, it can be observed that R can provide an indication of potential growth through elongation. For example, when the R-value is relatively small, it suggests that the glacier lake may be expanding in length.

Newly formed glacier lakes indeed have the potential to grow at a higher rate. This is because such lakes typically belong to the GCL type, receiving abundant glacier meltwater, which accelerates their rapid expansion. It is important to clarify that we do not determine whether a lake is newly formed based on the R-value, but rather through multi-temporal remote sensing imagery. Newly formed moraine-dammed lakes often exhibit relatively high R-values, as their length and width are not significantly different during early formation. As shown in Figure 5, GCL-1 lakes have an R-value of 0.75, which is much higher than that of other glacier lake types.

It would be useful to provide a general explanation of why the value of R would go from 0.1-0.6 in a GCL 2 lake to 0.5-1 when it gets detached from the glacier (GUL1) and the glacier continues to retreat. There is an example, but I am thinking as a general definition.

**Explanation and revision:** Thank you very much for your question. Regarding why the R-value of GCL 2 lakes ranges from 0.1 to 0.6, while that of GUL 1 lakes ranges from 0.5 to 1, these two categories are not directly comparable. Our model is designed by first classifying moraine-dammed lakes into two main categories, GCL and GUL,

with R-values ranging between 0 and 1 for each category. Therefore, the R-values can be compared within the same type of glacier lake.

In line 140 "through statistical analysis of glacial lake sizes for different types, we defined the threshold for R", which method and statistical significant of the values?

**Explanation and revision:** Thank you very much for your question. Regarding the determination of the R-value threshold, we first classified moraine-dammed lakes into two major categories, GCL and GUL, based on glacier lake inventory data. Then, we extracted the R-values for each category and performed descriptive statistics, including mean, median, mode, variance, and standard deviation. Based on the geometry of moraine-dammed lakes, we defined a threshold, such as 0.7 for GCL. Lakes with an R-value greater than or equal to 0.7 are categorized as GCL-1, while those with an R-value less than 0.7 are categorized as GCL-2. During the experimental process, this threshold of 0.7 was not determined in a single step but was finalized after multiple trials, comparing it with the geometry of moraine-dammed lakes.

Figure 3: A small figure with the axis direction would be appreciated, as would Figure 2, which is in a yz plane according to my interpretation.

**Explanation and revision:** Thank you very much for your suggestion. After considering the comments from other reviewers, we have revised Figure 2. Regarding your suggestion about the axis direction, I did not fully understand the specific meaning. Could you please clarify?

[Figure]

Line 199: if r=0 and n=0, m has to be m>0; so this sentence inline 200 "and in most cases, m is not equal to zero" makes no sense. If m =0, after indicating that r=0 and n=0, it means that there is no lake.

**Explanation and revision:** I sincerely apologize for the misunderstanding caused by my unclear description. In the vast majority of cases, m is greater than zero. However,

there are instances where it can be equal to 0. For example, the Lake Zhasuo Co (93.25°E, 30.31°N) in southeastern Tibet, m=n=0, because the surface morphology of this lake is rectangular. This point has been clarified in the paper (L203-206).

How do you account for the potential ice at the bottom of the lake?

**Explanation and revision:** Thank you very much for your question. My understanding is as follows: Some moraine-dammed lakes evolve from proglacial lakes (GCL), and before these lakes expand to their maximum extent controlled by the surrounding topography, the ice thickness at the glacier terminus remains relatively large and has not yet completely melted. In this scenario, ice may still exist at the lake bottom.

Does   figure 8-a axis y refer to errors?

**Explanation and revision:** Figure 8-a: The y-axis refers to the lake volume derived from the model.

Line 395 says, "The underwater landforms of some MDLs are not always completely flat." Are they ever flat?

**Explanation and revision:** Thank you for your question, which made me realize that my wording was not precise enough. What I intended to convey is that the bottom of moraine-dammed lakes is not always a smooth parabolic shape. Therefore, we have revised the sentence to: "The underwater landforms of some MDLs are not always a smooth parabolic shape."

Why Jialong Co and Bienong Co are representative of the other 42 lakes for the sensitivity analisys?

**Explanation and revision:** Because Jialong Co and Bienong Co as representatives of GUL and GCL of MDLs, respectively. Additionally, both moraine-dammed lakes are relatively large in size, and their water depths exceed 130 m.

---

## Author Comment (AC5)

Dear Prof. Emmer,

Thank you very much for your valuable comments and suggestions, which were crucial in improving the manuscript (MS). The MS has been extensively revised based on each point of the reviewers' comments and suggestions (see point-to-point responses below), and the updated sections of the revised MS have been highlighted.

Thanks again to the reviewers for their valuable comments and suggestions. We hope you find the revised MS, revision notes in order. If you have any questions, please do not hesitate to contact us (qmm@mail.ynu.edu.cn or shiyin.liu@ynu.edu.cn). Many thanks for your time and consideration.

**Response to Reviewer ,**

This study approximates glacial lake volume from simplified geometric representations of 4 main sub-types of moraine-dammed lakes, considering glacier-lake relationship (connected vs. unconnected) and lake width to length ratio. The performance of this new approach is reportedly better than the performance of other methods (comparison in Table 5).

**Explanation and revision:** Thank you very much for your review. Over the past few months, I have been on medical leave due to health reasons and have not been able to work. First, I would like to sincerely apologize for the delay in responding to your comments.

However, this is not surprising if the authors used the dataset of 44 Himalayan lakes with measured bathymetries to determine their parameters (section 3.3), and then use the same data to compare the performance of various methods (section 4.2). I hope I understood this correctly since the validation procedure is not described clearly in methods section. If I get it correctly, such performance evaluation is weak. A proper validation would require two independent datasets (training and testing).

**Explanation and revision:** Thank you very much for your questions and suggestions. The main reason for the misunderstanding is my lack of logical clarity and precision in wording, for which I sincerely apologize for the confusion caused.

First, regarding the determination of model input parameters, we have provided a detailed explanation in Section 3.3 of the paper. The unknown parameters in the model are w, l, a, m, n, and r, all of which can be extracted from glacier lake boundaries and DEM data. For example, we measured $w$ and $l$ by drawing a minimum rectangle bounding box with length $l$ encompassing the MDL (see Figure 4). To determine the slope a-value surrounding the MDL, we used a DEM with a spatial resolution of 12.5 m in the model computation. The detailed extraction steps can be found in Lines 230-238. Determining the appropriate thresholds for m, n, and r for different MDL types is challenging, as methods for extracting these parameters vary depending on the MDL types. In other words, due to the different types of glacial lakes, the values of m, n, and r vary. Additionally, these values change with the size of the glacial lake. To enable the model to automatically identify and calculate the corresponding m, n, and r for each glacial lake, we need to define a threshold. Based on the geometry of the glacial lake, we established a proportional relationship (Table 3) between $m$, $n$, $r$, and the glacier lake length ($l$). This proportional relationship is empirically defined and essentially represents a geometric segmentation of the glacial lake. The lake is divided into three sections, and the volume of each section is calculated separately. The total water storage of the lake is then obtained by summing the volumes of these three sections.

Therefore, we first used measured data from four glacial lakes to validate whether this proportional relationship was appropriate. After validation, we found that the empirically derived proportional relationship performed well. Hence, this study adopts this proportional relationship as the standard for the model's input parameters. No calibration or adjustments were made during this process. We have added the following explanation in lines 243 to 248 of the original text: " Based on the geometry

of the glacial lake, we established a proportional relationship between m, n, r, and the glacier lake length (l). This proportional relationship is empirically defined and essentially represents a geometric segmentation of the glacial lake. The lake is divided into three sections, and the volume of each section is calculated separately. The total water storage of the lake is then obtained by summing the volumes of these three sections."

To sum up, there is no parameter that needs to be trained or optimized, all parameters can be measured through the glacial lakes and their surrounding topology. The validation of the model is based on an independent dataset. The primary data and workflow for determining the parameters in the model are shown in the figure below.

[Figure]

Regarding the validation section you mentioned, I did not clearly explain the entire rationale and process for model validation. Therefore, based on the suggestions from

all reviewers, I have added a new subsection in the methodology section: 3.4. Model validation and application.

**3.4. Model validation and application**

In this study, we initially validated our parameterization using bathymetric measurements from four representative glacial lakes surveyed between 2020 and 2021. Subsequently, we combined the data from these four lakes with the remaining six glacier lakes we measured, along with water storage data from 34 MDLs obtained from relevant literature sources (see Appendix A for details). This resulted in a dataset of 44 lakes, which was used to compare and validate the performance of our model against other existing methods.

A glacier lake inventory of the High Mountain Asia region, published by Wang et al, 2020 was used as input data for the model application to assess the water storage of moraine-dammed lakes in this region. Notably, Wang's glacier lake inventory provides a detailed classification of GCL and GUL, which has been internationally recognized. It is important to note that in his dataset, GUL refers specifically to glacier lakes that do not contact glaciers, which may not necessarily all be moraine-dammed lakes. We conducted a thorough review and made revision to ensure that we retained only those GULs classified as moraine-dammed lake.

And the whole validation procedure is even more confusing since only 4 bathymetries are mentioned as input data for model validation in section 4.1. This is statistically not convincing, considering 4 sub-types of moraine-dammed lakes and number of parameters that are used. Further, a subset of 12 lakes is used in section 5.1 while 4 and 10 lakes are mentioned in Conclusions. This needs to be clarified.

**Explanation and revision:** Thank you very much for your review. We have added the main approach for model validation and comparison with other methods in Section 3.4 of the methodology. The four and ten lakes mentioned in the conclusion refer to the total of ten glacier lakes we measured in the field, of which four were used to validate the model's accuracy. These ten lakes, combined with 34 data points obtained

from the literature, formed a dataset used to compare and validate the effectiveness of our model against other methods.

The application section 4.3 is not linked to the methodology. It is not clear what was done and whether (and how?) all 13,166 lakes mapped by Wang et al. (2020) were classified according to the classification scheme used in this study and whether all these are moraine-dammed lakes?

**Explanation and revision:** Thank you very much for your review. We have added Section 3.4 of the methodology to link model application. In this study, we used Wang's data to estimate the water storage of moraine-dammed lakes in the High Mountain Asia region. His dataset has rigorously classified GCL and GUL. It is important to note that in his dataset, GUL refers specifically to glacier lakes that do not contact glaciers, which may not necessarily all be moraine-dammed lakes. We conducted a thorough review and made revision to ensure that we retained only those GULs classified as moraine-dammed lake.

Wang, X., Guo, X., Yang, C., Liu, Q., Wei, J., Zhang, Y., Liu, S., Zhang, Y., Jiang, Z., Tang, Z., 2020. Glacial lake inventory of high-mountain Asia in 1990 and 2018 derived from Landsat images. Earth System Science Data, 12(3), 2169-2182.

At the end, the importance of this improvement in lake volume estimation for GLOF studies (the main justification throughout the study) is unclear unless other (and much larger) sources of uncertainties in GLOF studies (e,g, coming up with realistic scenarios of GLOF triggers and GLOF mechanism, plausible breach development and dimensions, associated shape of the outburst hydrograph curve, % of lake volume release, etc.) are addressed.

**Explanation and revision:** Your understanding is indeed correct, and we share the same perspective. In my doctoral dissertation, we place significant emphasis on the flow processes at the dam breach, as well as the triggering factors for moraine-dammed lake outburst floods, such as ice falls, snow avalanches, and

landslides entering the lake, which generate waves and ultimately lead to the collapse of the dam. The following figures depict the reconstructed outburst flood flow process of the Cirenma Co (which experienced an outburst in southern Tibet in 1981), as well as schematic diagrams of external triggering factors for moraine-dammed lakes, such as ice falls, snow avalanches, and landslide hazard areas.

[Figure]

QI Miaomiao, LIU Shiyin, GAO Yongpeng, et al. Water volume changes and assessment of potential outburst triggers for glacial lakes in the Nidu Zangbo basin,

southeastern Tibet: a case study of Tanong Co［J］. Journal of Glaciology and Geocryology,2023,45(4):1205-1219.

L39-40: this definition is artificial; moraine-dammed lakes not only trap meltwater (how about water from liquid precipitation?); debris at or near the termini of glacier doesn't necessarily need to be a moraine

**Explanation and revision:** Thank you very much for your explanation. We have made appropriate revisions to the sentence, as shown below:

L39-40: "Moraine-dammed glacial lakes (MDLs) trap meltwater from snow, ice and liquid precipitation within basins behind dams at or near the termini of glaciers."

L53: ice- or landslide-dammed lakes may be unstable too

**Explanation and revision:** We apologize for the lack of clarity in our previous statement. Here, we revised the sentence as follows:

**L54-55:** "MDLs are prone to sudden failure due to the instability of the dam structure, releasing parts of the impounded water storage in catastrophic floods (Westoby et al., 2014)……"

L72-74: the peak discharge is rather linked to the magnitude of triggering event than lake volume

**Explanation and revision:** Thank you very much for your correction, our previous wording was not precise enough. In the revised manuscript, we have made the following adjustments:

**L74-77:** "The peak discharge during GLOFs is a commonly used parameter for assessing flood hazards and can be derived from empirical formulas related to the lake volume."

L77: how much was that?

**Explanation and revision:** The Sangwang Tsho experienced disastrous outbursts on July 16, 1954, with peak discharges of approximately 10,000 m³/s and a total lake volume of 71.6 × 10⁶ m³ (Patel et al., 2017; Veh et al., 2019). To ensure clarity, we have included specific values in the revised manuscript.

**L79-81:** "The Sangwang Tsho experienced disastrous outbursts in July 16, 1954, featuring one of the highest reported flood water storages ($71.6 \times 10^6$ m$^3$) and discharges ($\sim$10,000 m$^3 \cdot$s$^{-1}$) (Patel et al., 2017; Veh et al., 2019)……"

Patel, L.K., Sharma, P., Laluraj, C., Thamban, M., Singh, A., Ravindra, R., 2017. A geospatial analysis of Samudra Tapu and Gepang Gath glacial lakes in the Chandra Basin, Western Himalaya. Nat. Hazards 86, 1275–1290.

Veh G , Korup O , Walz A .Hazard from Himalayan Glacier Lake Outburst Floods[J].Proceedings of the National Academy of Sciences, 2019, 117(2).DOI:10.1073/pnas.1914898117.

L126-127: this indication is not clear since the ratio is dimensionless (really a width of 1 m?)

**Explanation and revision:** In our original text, the following description was provided: "According to the glacial lake inventory, the R value for glacial lakes in High Mountain Asia ranges from 0.1 to 1.0. When R is less than 0.1, it indicates the presence of glacial lakes with lengths exceeding 10 meters but widths of approximately 1 meter. **However, in reality, glacial lakes with such dimensions are practically non-existent.**" ……

Therefore, our intended meaning here is that, based on the glacier lake inventory data, the R-values fall between 0.1 and 1.0, which is an objective fact. Since glacier lakes with R-values less than 0.1 do not exist, the subsequent selection of thresholds for glacier lake classification based on R is set within the range of 0.1 to 1.0.

To avoid potential misunderstandings during the reading process, we have revised the sentence as follows:

**L130-132:** "If R is less than 0.1, it would indicate the presence of glacial lakes with lengths exceeding 10 meters but widths of approximately 1 meter. However, in reality, glacial lakes with such dimensions are practically non-existent……"

Fig. 2: please only display parameters that are further use (remove slope beta, points f and g)

**Explanation and revision:** We have revised the Figure 2.

[Figure]

Table 2: please clarify whether alpha is mean or median slope (as mentioned in Table 3); what is the influence of DEM acquisition date on alpha estimation?

**Explanation and revision:** I am very sorry for the misunderstanding caused by the lack of rigor in my expression. The slope here refers to the median slope (see figure below). The date of DEM acquisition has a certain influence on the slope, and the accuracy of dem also has an influence on the slope. However, considering that the degree of such influence is relatively small and the data of the same period are used in the assumptions of the modeling in this study, the influence is classified as the error of the model itself.

Table 4: what is simulated lake depth – a mean? And what do the two values in error column refer to?

**Explanation and revision:** The simulated water depth here refers to the mean depth. The data on the left side of the error column has not been updated, while the right side indicates the relative error between the simulated mean and the measured mean depth. In the revised manuscript, we carefully reviewed all the data and added relevant descriptions, as shown in the updated Table 4.

**Table 4** Validation results of the mathematical model.

| Name | Year of survey | Type | Area (km²) | Lake depth (m) | | | Water storage (10⁶ m³) | | |
| --- | --- | --- | --- | --- | --- | --- | --- | --- | --- |
| | | | | Observed (max/mean) | Simulated (mean) | Relative error | Observed | Simulated | Error |
| Bienong Co | 2021 | GCL2 | 1.16 | 181/74 | 109 | +47% | 102.00 | 95.689 | -6% |
| Maqiong Co | 2021 | GCL2 | 0.22 | 34/16 | 17 | +6% | 3.325 | 3.581 | +7% |
| Tanong Co | 2021 | GUL2 | 0.13 | 29/15 | 17 | +13% | 1.821 | 1.915 | +5% |
| Jialong Co | 2020 | GUL2 | 0.55 | 135/62 | 67 | +8% | 37.530 | 37.952 | +1% |

Table 5: some of the lakes (e.g. Imja Tsho or Jialong Co) are represented more than once. This may influence performance evaluation; the areas of Jialong Co do not match between Table 4 and 5)

**Explanation and revision:** Thank you very much for your thorough review; we are deeply impressed by your rigorous approach. Some lakes appear multiple times in Table 5 because they were measured by different teams in different years. We have included each instance as an independent data point in our sample set. In Tables 4 and 5, the area of Jialong Co measured by our team is 0.55 km², which I mistakenly

recorded due to an oversight. Thank you for pointing this out. We have made the correction and reviewed all the data.

L313-315: R^2 will always be very high (>0.95) for most of the methods

**Explanation and revision:** Judging solely from the goodness of fit between the model-derived data series and the measured data, the values are indeed not low. Therefore, this study employs multiple error evaluation methods to compare the accuracy of different approaches.

Figure 8: I don't understand what is the meaning of these box plots unless it is connected to measured data? The XY graph type (inset) is way more meaningful and the authors may consider showing a panel with performance of all methods in XY graphs.

**Explanation and revision:** I used box plots to represent the range of water storage estimates for each method, highlighting the differences between various approaches. Figure 8a shows significant variations among the methods, where it is clearly evident that some methods yield much larger estimates than our model, potentially resulting in order-of-magnitude discrepancies in the estimated water storage for individual lakes. The results from our model, by contrast, show a more concentrated distribution compared to the other methods. Figure 8b presents a comparison between our model's estimates and the measured values, specifically demonstrating the model's strong performance. We also experimented with other visualizations to compare the different methods, but overall, none were as effective as the box plot.

[Figure]

L339: the scaling up of the lake volume estimation procedure to the whole HMA is not properly described in methods.

**Explanation and revision:** We have carefully considered your suggestion and ultimately decided that no new content needs to be added to the methodology section. The main reasons are as follows: the primary focus of this study is the development of a new model, followed by validation of its accuracy using measured data, comparison with other methods, and finally, application of the new model for water volume estimation. Therefore, the estimation of water storage in glacial lakes across the High Mountain Asia region is conducted using the new model we developed. This has been explained in lines 348 to 349, as follows: "Therefore, this study employs our model to provide preliminary estimates of glacial lake water storages in the study area."

L367-376: this seems bit out of the context. Clearly, large lakes are frequently considered risky since lake area / volume is commonly used as GLOF susceptibility criteria.

**Explanation and revision:** Your concern has brought this issue to our attention. After considering the feedback from all reviewers, we have deleted this section in the revised manuscript to maintain coherence in the context.

~~L367-376: At least 88 MDLs had caused 122 lake outburst floods in this area before 2022 (Veh et al., 2019, 2022; Zheng et al., 2021a) (Figure 10a), constituting approximately 44% of the total GLOF count in High Mountain Asia. Zheng et al. (2021a) identified 280 MDLs within the study area with extremely high potential for outburst floods. Our model suggests that although the number of MDLs with a higher risk of outbursts is less than one-fifth of the total, their total water storage in 2022 exceeds 60% of the total water storage of MDLs in the study area. Furthermore, from 1990 to 2022, the total water storage of these high-risk MDLs increased from 2,019 ± 469 ×106 m3 to 5,622 ± 596 ×106 m3, representing a substantial growth of 178%, with an annual expansion rate of approximately 5.6%·a 1. This result is valuable as it enables practitioners to prioritize and focus their attention on areas where the largest flood water storages are expected.~~

L375: the annual expansion rate +5.6% a^-1 over 32 years does not correspond to a reported growth of 178% over this period

**Explanation and revision:** I sincerely apologize for my imprecise wording, which caused your misunderstanding. We have revised the sentence to: "Furthermore, from 1990 to 2022, the total water storage of these high-risk MDLs increased from $2,019 \pm 469 \times 10^6$ m$^3$ to $5,622 \pm 596 \times 10^6$ m$^3$, representing a substantial growth of 178%, with average annual expansion rate of approximately 5.6%·a$^{-1}$."

L395: they are not flat (as documented in your Fig. 10)

**Explanation and revision:** Thank you for your check, which made me realize that my wording was not precise enough. What I intended to convey is that the bottom of moraine-dammed lakes is not always a smooth parabolic shape. Therefore, we have revised the sentence to: "The underwater landforms of some MDLs are not always a smooth parabolic shape."

L433: what is MDLVL?

**Explanation and revision:** Thank you very much for pointing out the issue. We have reviewed and revised the entire text, changing "MDLVM" to "our model."

L453: the term "outburst water storage" is not appropriate. What is estimated here is a lake volume / lake water storage. It doesn't have much to do with outburst / outburst volume.

**Explanation and revision:** Thank you very much for your guidance. We have corrected the erroneous expression. Please review the revised sentence in Line : "Water storage plays a crucial role in predicting peak discharge of GLOFs."

To sum up, this study can help to improve moraine-dammed lake volume estimates in HMA. However, especially the validation process needs to be clarified and treated in statistically convincing way. I recommend major revisions.

**Explanation and revision:** Thank you very much for the opportunity to revise the manuscript. I have made point-by-point revisions in accordance with all the reviewers' comments and provided detailed clarifications on the areas where you had questions. Thank you again!

---

## Author Response (AR2)

**Dear Editor and Reviewers,**

Thank you very much for your valuable comments and suggestions, which were crucial in improving the manuscript (MS). The MS has been extensively revised based on each point of the reviewers' comments and suggestions (see point-to-point responses below), and the updated sections of the revised MS have been highlighted.

In addition, we would like to acknowledge the valuable guidance provided by Professor Zhifang Zhao during the revision process and in the preparation of the figures. As a result of his significant contributions, we have included him as the third author in this revision.

Thanks again to the reviewers for their valuable comments and suggestions. We hope you find the revised MS and revision notes in order. If you have any questions, please do not hesitate to contact us (qmm@mail.ynu.edu.cn or shiyin.liu@ynu.edu.cn). Many thanks for your time and consideration.

**Referee #1**

Thank you for detailed reply to my comments. Remaining issues: (i) I still think that the 4 bathymetries should not be used for both training and testing;

**Explanation and revision:** Thank you for your comment and for raising this concern. In this study, the four lakes (Bienong Co, Maqiong Co, Tanong Co, and Jialong Co) are not entirely used for training the model. Instead, they were employed to determine the buffer distance around the lakes. Our model for estimating water volume relies on several parameters, such as $w$, $m$, $n$, $r$, $l$, and $a$, which are manually defined rather than derived from training. Then, to validate the model's parameterization (Here, we are referring to the accuracy of the automated calculation of the model's input parameters $w$, $m$, $n$, $r$, $l$, and $a$), we tested it using bathymetric measurements from four representative glacial lakes surveyed between 2020 and 2021. These four lakes, along

with data from additional lakes, were then used as an independent sample set to evaluate and compare the performance of our model against other methods.

In summary, our model is not trained on these four lakes but is instead an idealized model defined based on field measurements and lake basin morphology. We appreciate your insightful feedback, which has allowed us to further clarify this point.

(ii) for the application part, it is written that "We conducted a thorough review and made revision to ensure that we retained only those GULs classified as moraine-dammed lake." - this procedure should be described in detail in the Methodology;

**Explanation and revision:** Thank you for your valuable suggestion regarding the description of our procedure. We have revised the text to include a detailed explanation of the methodology based on Yao et al.'s (2018) classification criteria for moraine-dammed lakes. The revised description clarifies our approach and ensures that the methodology is transparent and comprehensive. Please review the revised manuscript to check our updated description.

**Line 291-299**: *"It is important to note that in his dataset, GUL refers specifically to glacier lakes that do not contact glaciers, which may not necessarily all be moraine-dammed lakes. To ensure the accuracy of our analysis, we conducted a thorough review based on the classification criteria proposed by Yao et al., (2018) which identify three types of moraine-dammed lakes: (1) lakes situated between the end moraine ridge and the glacier terminus, (2) lakes beside the lateral moraine ridge, and (3) lakes on the moraine ridge. Each GUL in the dataset was individually assessed against these criteria, and only those meeting the classification as moraine-dammed lakes were retained for further analysis......"*

Yao, X. J., Liu, S. Y., Han, L., Sun, M. P., and Zhao, L. L.: Definition and classification system of glacial lake for inventory and hazards study. J. Geogr. Sci., 28, 229–241, https://doi.org/10.1007/s11442-018-1467-z, 2018.

(iii) please better highlight the importance of this work; the claim of more precise

prediction of a peak discharge is not supported by any example (the study only concerns the estimation of lake volume).

**Explanation and revision:** Thank you for your valuable feedback regarding the emphasis on the importance of our work. In response, we have revised the manuscript to better highlight the significance of estimating lake volume in the effective management of GLOF hazards. Specifically, we have removed the claim about the precise prediction of peak discharge and instead emphasized the critical role of understanding lake volume as a key factor in assessing GLOF likelihood and magnitude. We sincerely appreciate your thoughtful comments, which have helped us improve the clarity and focus of our study. Thank you for your efforts to strengthen our work. Please review the revised manuscript to check our updated description.

**Line 72-79:** *"Effective management of GLOF hazards hinges on the ability to assess both the likelihood and magnitude of such events (Clague et al., 2000). This typically requires understanding several critical factors, including the water storage of MDL, the structural integrity and stability of the dam, potential external triggers, and the flood's anticipated flow path (e.g., Richardson and Reynolds, 2000; Westoby et al., 2014; Mergili et al., 2020; Sattar et al., 2021). Estimating glacial lake volume, however, presents significant challenges. Many glacial lakes are situated in remote, physically demanding, and hazardous environments, complicating bathymetric surveys of the lake basins (Cook and Quincey, 2015; Qi et al., 2022)……"*

**Referee #2**

Thank you to the authors for clarifying the methodology and stating that the method presented applies geometry equations, the parameters of which are derived from observations. I have one last couple of questions:

-For the models, you need to estimate the depth using the angle of a buffer area around the lake. In equation 8, you use alpha, but in Table 2, 3, and Figure 6, you used "a". Do alpha and "a" represent the same parameter?

-If so, you should use the same, and it would be good to indicate which lakes were used to determine that 80 meters is a good buffer distance from the lake (Lines 232-240).

**Explanation and revision:** I sincerely apologize for any misunderstanding caused by my unclear explanation. Here, we used the parameter "$a$" to refer to slope angles of the glacial lake, they represent the same parameter. Please refer to Line 206 "Finally, the water depth ($h$) can be derived from the $w$ and slope angles ($a$) of the glacial lake" in the original text.

Based on your suggestion, we specified in the description that the lakes Bienong Co, Maqiong Co, Tanong Co, and Jialong Co were used to determine the optimal buffer distance.

**L232-234**: "*By comparing the simulated results with the measured data (lakes Bienong Co, Maqiong Co, Tanong Co, and Jialong Co), we found that the water storage estimation using the median value of a within 80 m external buffer zone had a lower relative error and higher overall accuracy…….*"

Regarding my comments about the axis, I suggested that indicating the direction of the axis in the figure would help to follow the article. However, after rereading that section, it now seems unnecessary.

**Explanation and revision:** Thank you for reassessing your comment on the axis direction. I'm glad the current presentation feels clear now. I truly appreciate your thoughtful feedback and efforts to improve the article.